# Selecting home care quality indicators based on the Resident Assessment Instrument-Home Care (RAI-HC) for Switzerland: A public health and healthcare providers' perspective

Aylin Wagner[1,2]*, Franziska Zúñiga[3], Peter Rüesch[1], René Schaffert[1‡], Julia Dratva[1,4‡], on behalf of the HCD Research Group[¶]

1 Institute of Health Sciences, School of Health Professions, ZHAW Zurich University of Applied Sciences, Winterthur, Switzerland, 2 Department of Health Sciences and Medicine, University of Lucerne, Lucerne, Switzerland, 3 Institute of Nursing Science, Department of Public Health, University of Basel, Basel, Switzerland, 4 Medical Faculty, University of Basel, Basel, Switzerland

‡ These authors share last authorship on this work.
¶ Membership of the HCD Research Group is provided in the Acknowledgments.
* aylin.wagner@zhaw.ch

**Data Availability Statement:** The data underlying the findings are available on Harvard Dataverse:

## Abstract

### Background

Despite an increasing importance of home care, quality assurance in this healthcare sector in Switzerland is hardly established. In 2010, Swiss home care quality indicators (QIs) based on the Resident Assessment Instrument-Home Care (RAI-HC) were developed. However, these QIs have not been revised since, although internationally new RAI-HC QIs have emerged. The objective of this study was to assess the appropriateness of RAI-HC QIs to measure quality of home care in Switzerland from a public health and healthcare providers' perspective.

### Methods

First, the appropriateness of RAI-HC QIs, identified in a recent systematic review, was assessed by a multidisciplinary expert panel based on the RAND/UCLA Appropriateness Method taking into account indicators' public health relevance, potential of influence, and comprehensibility. Second, the QIs selected by the experts were afterwards rated regarding their relevance, potential of influence, and practicability from a healthcare providers' perspective in focus groups with home care nurses based on the Nominal-Group-Technique. Data were analyzed using median scores and the Disagreement Index.

### Results

18 of 43 RAI-HC QIs were rated appropriate by the experts from a public health perspective. The 18 QIs cover clinical, psychosocial, functional and service use aspects. Seven of the 18 QIs were subsequently rated appropriate by home care nurses from a healthcare providers' perspective. The focus of these QIs is narrow, because three of seven QIs are pain-related. From both perspectives, the majority of RAI-HC QIs were rated inappropriate because of insufficient potential of influence, with healthcare providers rating them more critically.

Wagner, Aylin, 2020, "Expert panel (RAM) responses and results of rating round 1 and 2 - Selecting home care quality indicators based on the Resident Assessment Instrument-Home Care (RAI-HC) for Switzerland - a public health and healthcare providers' perspective. https://doi.org/10.7910/DVN/3BDYX6 Wagner, Aylin, 2020, "Focus groups (NGT) responses and results - Selecting home care quality indicators based on the Resident Assessment Instrument-Home Care (RAI-HC) for Switzerland - a public health and healthcare providers' perspective". https://doi.org/10.7910/DVN/WUVUII

**Funding:** This study was funded by the Swiss National Science Foundation (SNSF), National Research Program 74 "Smarter Health Care", Project "Swiss Home Care Data: patient profiles and quality measures for home care" (No. 167499). The funder had no role in study design, data collection and analysis, decision to publish, or preparation of the manuscript.

**Competing interests:** The authors have declared that no competing interests exist.

**Abbreviations:** DI, Disagreement Index; FOPH, The Federal Office of Public Health; GP, General practitioner; NGT, Nominal-Group-Technique; QI, Quality indicator; RAI-HC, Resident Assessment Instrument-Home Care; RAM, RAND/UCLA Appropriateness Method; UI, Urinary incontinence.

## Conclusions

The study shows that the appropriateness of RAI-HC QIs differs according to the stakeholder perspective and the intended use of QIs. The findings of this study can guide policymakers and home care organizations on selecting QIs and to critically reflect on their appropriate use.

## Background

The ageing of the population and increase in life expectancy is associated with a growing number of people with one or more chronic conditions, leading to a higher demand of home care [1]. Home care supports patient's rehabilitation process and can help sustain their independence and, thus, meet the desire of the majority of older adults to remain in their own home for as long as possible [2]. Home care services in Switzerland are intended for people of all age groups in need of care or assistance at home and are run by profit and non-profit home care organizations as well as independent nurses. Four-fifths of Swiss home care clients receive services from non-profit home care organizations [3]. The range of services offered by home care organizations includes nursing care and domestic tasks [4]. The compulsory health insurance pays for care services prescribed by general practitioners (GPs) but not for domestic tasks. The organizational structure of home care is highly decentralized and ultimately reflects the federal political structure of Switzerland. Home care plays an important role in managing interfaces between primary care, acute care, long-term care and mental health services [5] and is characterized by interprofessional collaboration, i.e. nurses, GPs and other health care providers (e.g. pharmacists, physiotherapist) work together to provide a wide range of services to clients [6, 7]. However, despite the increasing importance of home care, quality assurance in this health care sector is hardly established in Switzerland, in contrast to other sectors such as hospitals [8].

The Institute of Medicine (IOM) defines quality of care as the degree to which health services for individuals and populations increase the likelihood of desired health outcomes and are consistent with current professional knowledge [9]. Various stakeholders such as healthcare providers, policy-makers and patients have different perspectives from which quality of care can be viewed. The perspective of healthcare providers, for example, focuses primarily on the care provided to individual patients [10, 11]. The public health perspective, on the other hand, tends to place more weight on population health and the functioning of health care systems [10]. The different perspectives and priorities of stakeholders must be considered when assessing quality of care [12, 13].

Health care quality can be assessed, monitored and evaluated with quality indicators (QIs) [14]. In order to measure quality meaningfully, it is important that QIs meet certain quality requirements. They must be relevant to the selected problem and field of application, feasible, valid, reliable, influenceable, understandable, and sensitive to change [15–17]. The development of QIs can be divided into two phases. First, the identification of candidate QIs and the corresponding scientific evidence, and second, the QI assessment, consisting of panel review, risk adjustment and empirical analysis [15–19]. Because scientific evidence on QIs is often limited, it is necessary to combine available evidence with expert opinion using Delphi techniques [16].

The international research consortium interRAI has developed home care QIs based on data collected with the Resident Assessment Instrument-Home Care (RAI-HC or interRAI-HC) [5]. RAI-HC is a standardized assessment tool and care planning instrument for

long-stay home care clients adopted by home care organizations in several countries [20]. InterRAI developed the first RAI-HC QI set in 2004 [5] and a second, updated QI set in 2013 [20]. The RAI-HC QIs are constructed as proportions or percentages, expressed by a fractional calculation with numerator (number of clients with a particular outcome) and denominator (number of clients at risk for the outcome and not otherwise excluded from the QI) [5, 21]. A systematic review showed that currently 48 RAI-HC QIs exist [22]. These QIs cover different areas relevant to home care, focusing on functional (e.g. activities of daily living, cognition, communication, hearing, eyesight), clinical (e.g. bladder incontinence, bowel incontinence, skin ulcer, mouth problems, falls, weight, mood, pain), social (informal caregivers, social isolation), and service use aspects (flu vaccination, hospitalization) [22].

In Switzerland, an adapted and shorter version of the original and internationally used English-language interRAI-HC has been implemented in 2003 for use in all home care organizations [23]. Based on the Swiss RAI-HC and the first interRAI QI set from 2004 [5], RAI-HC QIs were developed for Switzerland in 2010 [24]. These Swiss RAI-HC QIs have not been reviewed and revised since their implementation, although new international RAI-HC QIs [20] have emerged in the meantime. The Swiss RAI-HC QIs have so far only been used for internal quality management in non-profit home care organizations. To date, in Switzerland, RAI-HC QIs (or any home care QIs) are not reported and there are no national standards for home care [8]. However, there is a legal basis that obliges home care organizations to report data on QIs to the respective federal authorities with a goal of public reporting [25]. Currently this law is not being implemented due to lack of knowledge which QIs are the most appropriate. The Federal Office of Public Health (FOPH) will define which QIs will be collected at the national level in near future. No incentives will be linked to the QI reporting.

The aim of this study was to assess the appropriateness of RAI-HC QIs to measure quality of home care in Switzerland from a public health and healthcare providers' perspective based on a consensus approach. The study is a subproject of the study "Better data on the quality of home care", which aims to expand the Swiss RAI-HC data and to explore its research potential in the field of home care and long-term care.

## Methods

We chose a grounded consensus, two-phase approach to assess the appropriateness of RAI-HC QIs identified by the authors in a recent systematic literature review [22].

In phase 1, we conducted an expert panel using the RAND/UCLA Appropriateness Method (RAM) [26] and the proposed standards for Guidance on Conducting and Reporting Delphi Studies (CREDES) (S1 File) [27]. For the expert panel rating, five of the 48 identified RAI-HC QIs [22] were excluded because the respective QIs were not calculable with the Swiss version of the RAI-HC and therefore not applicable in the Swiss context. The experts rated the appropriateness of the remaining RAI-HC QIs to measure home care quality in Switzerland from a public health perspective.

In phase 2, we held focus groups with home care nurses from various Swiss home care organizations following the Nominal-Group-Technique (NGT) [28]. In this second phase, the healthcare providers evaluated the appropriateness of RAI-HC QIs rated to be appropriate by the experts from their practical perspective. Fig 1 visualizes the two-phase rating and selection process.

### Ethical considerations

The study was submitted to the Cantonal Ethics Committee of the Canton of Zurich, Switzerland. The study does not fall under the Human Research Act and an exemption of an ethical

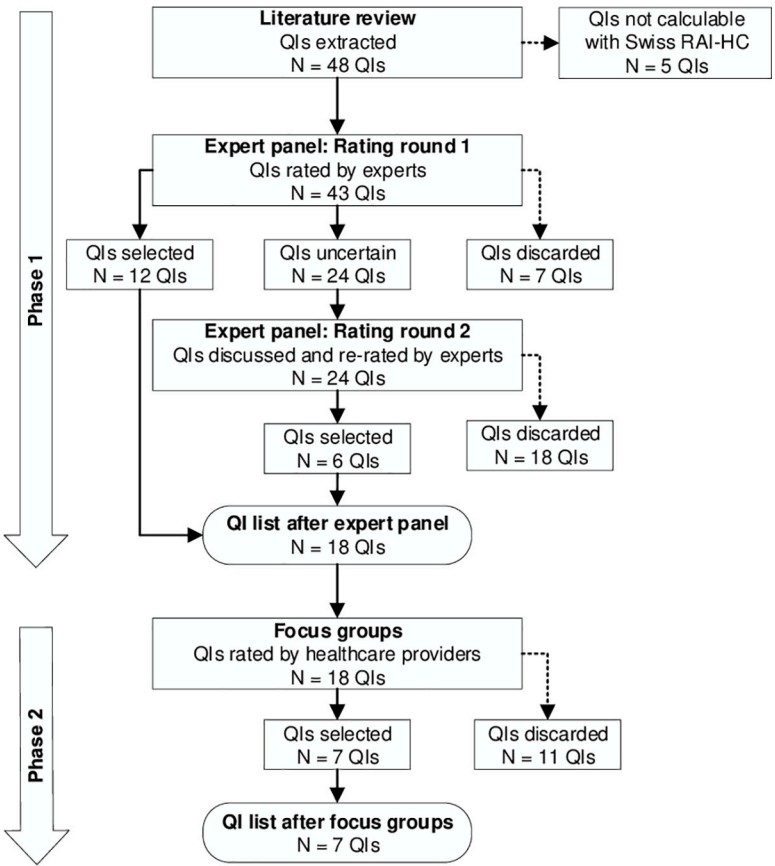

**Fig 1. Two-phase rating and selection process of quality indicators.** QIs, Quality indicators; RAI-HC, Resident Assessment Instrument-Home Care.

review was received. The participants in the expert panel and the focus groups provided written informed consent to participate in the study. They were asked for permission to audiotape the expert panel meeting and the focus groups, informed of the assurance of participant anonymity, and how the data would be analysed and published.

## Phase 1: Expert panel (RAM)

The RAM is a modified Delphi technique developed in the 1980s by the RAND Corporation and the University of California Los Angeles and has been incorporated into a comprehensive approach for the development of QIs in various contexts [29–33]. The method systematically combines scientific evidence and expert opinion by asking experts to rate, discuss and re-rate QIs. This includes several steps, starting with a systematic review of the available evidence and the extraction of candidate QIs, a first individual rating round, then a face-to-face panel meeting with discussion and a second individual rating round [26].

**Expert panel composition.** The multidisciplinary expert panel consisted of 14 members from three language regions of Switzerland in order to take cultural differences into account and to examine the appropriateness of RAI-HC QIs throughout the country. The experts with various professional backgrounds had solid knowledge and professional experience in quality management and home care. Experts whose mother tongue was not German had very good passive language skills in German. For the composition of the panel see Table 1. Panel

**Table 1. Expert panel members.**

| Representatives from: | N |
|---|---|
| Cantonal Departments of Health | 4 |
| Organizations focusing on patient safety and quality in health care | 3 |
| Management of home care organizations | 3 |
| Home care or nursing associations | 2 |
| University of Applied Sciences related to public health | 1 |
| Federal Office of Public Health Switzerland (FOPH) | 1 |

N, number of experts.

members were selected based on their experience and expertise in the field of home care and healthcare quality. Efforts were also made to include representatives from the various language regions of Switzerland. A total of 18 experts recommended by the research team were invited by e-mail to participate in the study.

**Rating round 1.** Panel members received a list of 43 candidate RAI-HC QIs and a summary of the literature review (QI definitions with numerator and denominator, evidence on validity and reliability) [22], rating instructions and a description of the study method to ensure that the experts had access to the same body of evidence and information.

Panel members were asked to rate each RAI-HC QI for the appropriateness to measure quality of home care in Switzerland taking into account public health relevance, potential of influence, and comprehensibility. Table 2 describes the rating criteria in detail. The panel members could also suggest additional QIs or quality areas not yet covered by the candidate QIs.

**Analysis.** The analysis was carried out in accordance with the RAND/UCLA Appropriateness Method user's manual [26]. Median scores as well as level of agreement among panel members were calculated. Median scores in the range of 1–3 were classified as inappropriate, 3.5–6 as neither inappropriate nor appropriate (uncertain result), and 6.5–9 as appropriate. Level of agreement was assessed with the Disagreement Index (DI). The DI is based on the dispersion of the distribution (interpercentile range, IPR) and symmetry (interpercentile range adjusted for symmetry, IPRAS) of the ratings on the 9-point scale and is calculated with the formula: IPR (difference between 30th and 70th percentile) divided by the IPRAS. DI > 1 indicates a lack of consensus and DI ≤ 1 consensus among panel members [26]. Based on the

**Table 2. Rating criteria for expert panel and focus groups.**

| Rating criteria | Public health perspective (expert panel) | Healthcare providers' perspective (focus groups) | Answer options[a] |
|---|---|---|---|
| **Relevance** | The relevance of the QI for the Swiss health care system, i.e. relevance to improve home care and health of the home care population. | The relevance of the QI for the quality of home care, i.e. relevance to improve home care and health of a home care client. | 9-point scale |
| **Potential of influence** | The potential to influence the outcome (e.g. pain) measured by the QI through actions of the home care organization (e.g. management, home care nurses). | The potential to influence the outcome (e.g. pain) measured by the QI through actions of healthcare providers, i.e. home care nurses. | 9-point scale |
| **Comprehensibility** | The comprehensibility of the QI, i.e. definition is understandable. | Not asked | Yes-no |
| **Practicability** | Not asked | Reliability of RAI-HC items used for QI calculation, i.e. can the items be reliable coded by home care nurses. | 9-point scale |

QI, Quality indicator.

[a]9-point scale: 1 = lowest score, 9 = highest score.

median scores for relevance and potential of influence and the DI, we classified the QIs as selected, discarded or uncertain (see Table 3 for exact classification rules). Only uncertain QIs were later discussed in the expert panel meeting and re-rated.

Some of the 43 RAI-HC QIs taken from the systematic review [22] related to the same health outcome and represented alternative formulations (e.g. decline, improvement). We identified such QIs for six health outcomes: bladder continence, cognition, communication, activities of daily living, instrumental activities of daily living, and mood. All of the related QIs were classified as uncertain in the first rating round, should one of the QIs in question be classified as uncertain or discarded, and were thus discussed and re-rated with respect to the criterion potential of influence in the panel meeting.

**Panel meeting and rating round 2.** The panel members attended a face-to-face multilingual meeting (i.e. the experts could talk in their first language), led by an experienced moderator. Panel members were provided with a copy of the results of the first rating round, including their own individual rating results and a summary of the group ratings with median scores and level of agreement (DI). Individual ratings of other panel members were not revealed. During the meeting, the experts discussed the QIs classified as uncertain in the first rating round and subsequently individually re-rated the QIs. The discussion focused on evidence supporting the decision to select or decline the QIs.

The selection of the QIs followed the same rules as in the first rating round (see Table 3), i.e. QIs with appropriate median scores of relevance and potential of influence, and consensus, were selected. QIs which did not meet these criteria were discarded.

## Phase 2: Focus groups with healthcare providers

The QIs selected in the expert panel were further evaluated based on the Nominal-Group-Technique (NGT) [28] in three focus groups with registered nurses from Swiss home care organizations situated in the three main language regions. The documents for the focus groups were professionally translated from German into French, and focus groups were held in the respective language.

The participants were recruited with the support of the umbrella organization of non-profit home care organizations, Spitex Schweiz, and invited by e-mail to participate in the focus groups. All participants had several years of professional experience in home care and in the application of RAI-HC. The aim of the focus groups was to obtain knowledge whether the QIs rated as appropriate from a public health perspective by the experts were also suitable from the perspective of healthcare providers.

**Table 3. Classification rules of quality indicators.**

| Categories | Expert panel | Focus groups |
|---|---|---|
| | **Classification rules** | **Classification rules** |
| **Selected** | If median scores of **relevance** $\geq$ **6.5** (with DI $\leq$ 1) and **potential of influence** $\geq$ **6.5** (with DI $\leq$ 1) | If median scores of **relevance** $\geq$ **6.5** (with DI $\leq$ 1) and **potential of influence** $\geq$ **6.5** (with DI $\leq$ 1) and **practicability** $\geq$ **6.5** (with DI $\leq$ 1) |
| **Discarded** | If median scores of **relevance 3.5–6** (with DI $\leq$ 1 or > 1) and **potential of influence 3.5–6** (with DI $\leq$ 1 or > 1) | If median scores of **relevance** < **6.5** (with DI $\leq$ 1 or > 1) and/or **potential of influence** < **6.5** (with D I $\leq$ 1 or > 1) and/or **practicability** < **6.5** (with DI $\leq$ 1 or > 1) |
| | Or | |
| | If median scores of **relevance** $\leq$ **3** (with DI $\leq$ 1 or > 1) and/or **potential of influence** $\leq$ **3** (with DI $\leq$ 1 or > 1) | |
| **Uncertain** | If median scores of **relevance** $\geq$ **6.5** (with DI $\leq$ 1 or > 1) and **potential of influence 3.5–6** (with DI $\leq$ 1 or > 1) | Not applicable |

DI, Disagreement Index.

The NGT [28] is a structured consensus process and is based on a strongly structured meeting in which individual and group work alternate. Using this technique, participants independently rated the QIs at the beginning (first rating round) and at the end of the focus group (second rating round). The method allowed the research team to provide oral explanations on the QIs and to help participants in case of uncertainties during the rating process, as the home care nurses had no expertise in QI construction and use. Participants were asked to rate the QIs for the appropriateness to measure quality of home care taking into account healthcare providers' relevance, potential of influence, and practicability (see Table 2). Between the two rating rounds, the ratings for each QI were collected and followed by a discussion in which participants described the rationale of their ratings.

For the data analysis, the ratings of the second rating rounds of the three focus groups were merged. As in the expert panel, median scores and level of agreement (DI) among participants were calculated and the rule for QI selection was applied (see Table 3), i.e. only QIs with appropriate median scores of relevance, potential of influence, practicability, and consensus, were selected.

## Results

### Phase 1: Expert panel (RAM)

Based on the median scores and level of agreement (DI) from the first rating round, 12 QIs were selected, seven QIs were discarded and 24 QIs were classified as uncertain. The proportion of yes-responses for the criterion comprehensibility was $\geq$ 79% for all 43 QIs, i.e. all QIs were rated as comprehensible by the experts. The experts suggested further quality topics of relevance such as process of care, patient satisfaction, and quality of life. The investigators evaluated the suggestions and concluded that based on the currently available Swiss RAI-HC data developing and calculating such QIs was not possible.

13 panel members attended the panel meeting and re-rated the QIs. Based on the median scores and level of agreement (DI) of the second rating, six QIs were selected and 18 QIs were discarded. The final list consists of 18 QIs rated by the experts in the first or second rating as appropriate to measure quality of home care, taking into account public health relevance, potential of influence, and comprehensibility. The majority of QIs were discarded because of inappropriate rating results with respect to the criterion potential of influence. Table 4 shows the rating results for each rating round and indicates which QIs were selected by the expert panel.

### Phase 2: Focus groups with healthcare providers

The 18 RAI-HC QIs rated as appropriate in the expert panel were discussed and evaluated in three focus groups with registered nurses from Swiss home care organizations. Two focus groups were held in the German speaking part of Switzerland with ten and nine participants, respectively, and one focus group in the French speaking part of Switzerland with six participants, one of them representing the Italian speaking part of Switzerland.

Table 5 shows the rating results of the focus groups and the QI selection. 16 QIs were rated as appropriate from the healthcare providers' perspective with respect to relevance, seven QIs with respect to potential of influence and 12 QIs with respect to practicability. Only for one QI, the focus group found no consensus (DI > 1). Based on the overall result for the three rating criteria, seven QIs were selected and 11 QIs were discarded.

**Table 4. Public health expert panel ratings (RAND/UCLA Appropriateness Method).**

| Quality indicator[a] | Quality indicator characteristics | | | Rating round 1[e] | | | | | | Rating round 2[f] | | | | |
|---|---|---|---|---|---|---|---|---|---|---|---|---|---|---|
| | | | | Relevance | | Potential of influence | | Comprehensibility | Result | Relevance | | Potential of influence | | Final result |
| | Measure level[b] | Type[c] | Set[d] | Median | DI | Median | DI | Proportion of yes-response (in %) | | Median | DI | Median | DI | |
| Inadequate pain control | O | P | interRAI 1st | 9 | 0.1 | 7 | 0.3 | 100 | selected | . | . | . | . | selected |
| Improvement of pain | O | I | interRAI 2nd | 9 | 0.1 | 7 | 0.2 | 100 | selected | . | . | . | . | selected |
| Daily severe pain | O | P | interRAI 1st | 8.5 | 0.2 | 7 | 0.3 | 92 | selected | . | . | . | . | selected |
| Dehydration | O | P | interRAI 1st | 8 | 0.3 | 8 | 0.4 | 92 | selected | . | . | . | . | selected |
| Inconsistent drug intake | O | P | Swiss RAI-HC | 8 | 0.1 | 8 | 0.1 | 93 | selected | . | . | . | . | selected |
| Bladder continence (decline)* | O | I | interRAI 1st | 8 | 0.3 | 7 | 0.4 | 92 | uncertain (selected*) | . | . | 7 | 0.2 | selected |
| Delirium | O | P | interRAI 1st | 8 | 0.1 | 7 | 1.0 | 79 | selected | . | . | . | . | selected |
| Social isolation with distress | O | P | interRAI 1st | 8 | 0.3 | 6 | 0.4 | 100 | uncertain | . | . | 7 | 0.5 | selected |
| Informal caregiver distress | O | P | Swiss RAI-HC | 8 | 0.3 | 6 | 0.5 | 86 | uncertain | . | . | 7 | 0.1 | selected |
| Decline independency | O | P | Swiss RAI-HC | 8 | 0.2 | 7 | 0.5 | 85 | selected | . | . | . | . | selected |
| Skin ulcer | O | I | interRAI 1st | 7.5 | 0.3 | 7 | 0.2 | 92 | selected | . | . | . | . | selected |
| Obstipation | O | I | Swiss RAI-HC | 7 | 0.6 | 7 | 0.7 | 100 | selected | . | . | . | . | selected |
| Rehabilitation potential and no therapies | P | P | interRAI 1st | 7 | 0.4 | 7 | 0.6 | 50 | selected | . | . | . | . | selected |
| Difficulty in locomotion and no assistive device | O | P | interRAI 1st | 7 | 1.3 | 7 | 0.7 | 86 | uncertain | 7 | 0.5 | . | . | selected |
| Impaired locomotion in home | O | I | interRAI 1st | 7 | 0.5 | 6 | 0.7 | 93 | uncertain | . | . | 7 | 0.1 | selected |
| Hospitalization, ED, emergent care | O | P | interRAI 1st | 7 | 0.6 | 6 | 0.5 | 93 | uncertain | . | . | 7 | 0.4 | selected |
| Mouth problems | O | P | Swiss RAI-HC | 7 | 0.4 | 7 | 0.2 | 100 | selected | . | . | . | . | selected |
| Neglect or abuse | O | P | interRAI 1st | 7 | 0.2 | 6.5 | 0.8 | 93 | selected | . | . | . | . | selected |
| Cognitive function (decline or no improvement)* | O | I | interRAI 1st | 8 | 0.3 | 5 | 0.9 | 79 | uncertain | . | . | 6 | 0.9 | discarded |
| Cognitive function (decline)* | O | I | interRAI 2nd | 8 | 0.3 | 5 | 0.9 | 93 | uncertain | . | . | 6 | 1.7 | discarded |
| Unintended weight loss (measured with BMI) | O | P | interRAI 1st | 8 | 0.5 | 6 | 0.8 | 100 | uncertain | . | . | 6 | 0.8 | discarded |
| Falls | O | P | interRAI 1st | 8 | 0.3 | 6 | 0.5 | 86 | uncertain | . | . | 6 | 0.4 | discarded |
| ADL (decline)* | O | I | interRAI 2nd | 8 | 0.4 | 6 | 0.5 | 93 | uncertain | . | . | 4.5 | 1.0 | discarded |
| ADL (improvement)* | O | I | interRAI 2nd | 8 | 0.4 | 7 | 0.6 | 100 | uncertain (selected*) | . | . | 4.5 | 0.9 | discarded |
| Cognitive function (improvement)* | O | I | interRAI 2nd | 8 | 0.5 | 5 | 0.9 | 93 | uncertain | . | . | 3.5 | 0.4 | discarded |
| Prevalence of negative mood* | O | P | interRAI 1st | 7.5 | 0.7 | 5 | 0.6 | 100 | uncertain | . | . | 6 | 0.2 | discarded |
| Negative mood (improvement)* | O | I | interRAI 2nd | 7 | 0.4 | 5.5 | 0.5 | 100 | uncertain | . | . | 6 | 0.5 | discarded |
| Bladder continence (improvement)* | O | I | interRAI 2nd | 7 | 0.2 | 7 | 0.4 | 100 | uncertain (selected*) | . | . | 6 | 0.5 | discarded |

*(Continued)*

**Table 4.** (Continued)

| Quality indicator[a] | Quality indicator characteristics | | | Rating round 1[e] | | | | | | Rating round 2[f] | | | | |
|---|---|---|---|---|---|---|---|---|---|---|---|---|---|---|
| | | | | Relevance | | Potential of influence | | Comprehensibility | Result | Relevance | | Potential of influence | | Final result |
| | Measure level[b] | Type[c] | Set[d] | Median | DI | Median | DI | Proportion of yes-response (in %) | | Median | DI | Median | DI | |
| IADL (decline or no improvement)* | O | I | Swiss RAI-HC | 7 | 0.5 | 7 | 0.6 | 79 | uncertain | . | . | 5 | 1.5 | discarded |
| Negative mood (decline)* | O | I | interRAI 2nd | 7 | 0.3 | 5 | 0.9 | 100 | uncertain | . | . | 5 | 0.9 | discarded |
| IADL (decline)* | O | I | interRAI 2nd | 7 | 0.5 | 7 | 1.0 | 100 | uncertain *(selected*)* | . | . | 4.5 | 1.0 | discarded |
| Unfavorable weight change (measured with BMI) | O | I | Swiss RAI-HC | 7 | 0.4 | 6 | 1.0 | 92 | uncertain | . | . | 4 | 0.5 | discarded |
| IADL (improvement)* | O | I | interRAI 2nd | 7 | 0.5 | 5 | 0.5 | 100 | uncertain | . | . | 3 | 0.5 | discarded |
| Does not go out but used to | O | P | interRAI 2nd | 7 | 0.4 | 6 | 0.0 | 93 | uncertain | . | . | 3 | 0.8 | discarded |
| Hearing impairment | O | P | Swiss RAI-HC | 7 | 0.7 | 3 | 1.0 | 100 | discarded | . | . | . | . | discarded |
| Eyesight impairment | O | P | Swiss RAI-HC | 7 | 1.5 | 3 | 1.5 | 86 | discarded | . | . | . | . | discarded |
| ADL (decline or no improvement)* | O | I | interRAI 1st | 6.5 | 0.7 | 6 | 0.5 | 86 | uncertain | . | . | 5 | 1.0 | discarded |
| Bladder continence (decline, updated version)* | O | I | interRAI 2nd | 6 | 0.5 | 5.5 | 1.3 | 92 | uncertain *(discarded*)* | . | . | 7 | 0.9 | discarded |
| Communication (decline or no improvement)* | O | I | interRAI 1st | 6 | 0.7 | 5 | 1.0 | 79 | discarded | . | . | . | . | discarded |
| No desired weight change (measured with BMI) | O | I | Swiss RAI-HC | 6 | 0.3 | 5 | 1.8 | 92 | discarded | . | . | . | . | discarded |
| Communication (decline)* | O | I | interRAI 2nd | 6 | 0.8 | 4.5 | 0.7 | 93 | discarded | . | . | . | . | discarded |
| Communication (improvement)* | O | I | interRAI 2nd | 6 | 0.9 | 4.5 | 0.7 | 93 | discarded | . | . | . | . | discarded |
| Bowel incontinence | O | I | Swiss RAI-HC | 5.5 | 1.0 | 4 | 0.7 | 92 | discarded | . | . | . | . | discarded |

ADL, Activities of daily living; BMI, Body mass index; DI, Disagreement Index; ED, Emergency department; IADL, Instrumental activities of daily living; QI, Quality indicator.

*Identifies QIs that measure the same health outcome and represent alternative formulations. If one of three QIs related to the same health outcome were classified as uncertain or discarded in rating round 1, then all three QIs were classified as uncertain (regardless of the actual rating result) and were re-rated according to the criterion potential of influence in rating round 2.

The actual rating result of rating round 1 is indicated in the result column in italics, parentheses and marked by an asterisk (*).

[a]QIs identified in a systematic literature review [22].

[b]Measure level: O = Outcome, P = Process; classified by authors.

[c]Type: I = Incidence measure (measures changes in a client's health status from one time point to another), P = Prevalence measure (measures client's health status at a single point in time).

[d]QI set (origin of the QI): interRAI 1st = interRAI's 1st generation QI set developed in 2004 [5]; interRAI 2nd = interRAI's 2nd generation QI set developed in 2013 [20]; Swiss RAI-HC = Swiss RAI-HC QI set developed in 2010 [24].

[e]Median: Scores on a 9-point scale, 1 = lowest score, 9 = highest score.

DI: DI $\leq$ 1 means no extreme variation and indicates agreement, DI $>$ 1 means extreme variation and indicates disagreement.

Rating criteria: Relevance = The relevance of the QI for the Swiss health care system; Potential of influence = The potential to influence the outcome measured by the QI through actions of the home care organization (e.g. management, home care nurses); Comprehensibility = QI definition is understandable.

[f]. = not discussed and re-rated in rating round 2.

**Table 5. Healthcare provider focus group ratings.**

| Quality indicator | Relevance | | Potential of influence | | Practicability | | Result |
|---|---|---|---|---|---|---|---|
| | Median | DI | Median | DI | Median | DI | |
| Daily severe pain | 9 | 0.1 | 7 | 0.2 | 8 | 0.2 | selected |
| Skin ulcer | 8.5 | 0.1 | 8 | 0.2 | 9 | 0.1 | selected |
| Improvement of pain | 8 | 0.1 | 7 | 0.2 | 8 | 0.2 | selected |
| Obstipation | 8 | 0.2 | 7 | 0.2 | 7 | 0.4 | selected |
| Inadequate pain control | 8 | 0.1 | 7 | 0.4 | 7 | 0.2 | selected |
| Informal caregiver distress | 8 | 0.0 | 7 | 0.4 | 7 | 0.4 | selected |
| Dehydration | 8 | 0.0 | 7 | 0.4 | 6.5 | 0.5 | selected |
| Mouth problems | 8 | 0.2 | 6 | 0.5 | 7 | 0.4 | discarded |
| Inconsistent drug intake | 8 | 0.4 | 6 | 0.3 | 6 | 0.5 | discarded |
| Hospitalization, ED, emergent care | 8 | 0.3 | 5 | 0.3 | 8 | 0.3 | discarded |
| Difficulty in locomotion and no assistive device | 7 | 0.2 | 6 | 0.3 | 8 | 0.2 | discarded |
| Bladder continence (decline) | 7 | 0.4 | 6 | 0.5 | 6 | 0.5 | discarded |
| Social isolation with distress | 7 | 0.4 | 5 | 0.9 | 7 | 0.3 | discarded |
| Impaired locomotion in home | 7 | 0.6 | 5 | 0.6 | 7 | 0.4 | discarded |
| Delirium | 7 | 0.7 | 5 | 0.6 | 6 | 0.6 | discarded |
| Decline independency | 7 | 0.7 | 5 | 0.9 | 5 | 1.0 | discarded |
| Rehabilitation potential and no therapies | 6 | 0.7 | 5 | 1.0 | 4 | 1.0 | discarded |
| Neglect or abuse | 5 | 1.6 | 5 | 0.9 | 5 | 1.7 | discarded |

DI, Disagreement Index; ED, Emergency department.

DI: DI ≤ 1 means no extreme variation and indicates agreement, DI > 1 means extreme variation and indicates disagreement.

Median: Scores on a 9-point scale, 1 = lowest score, 9 = highest score.

Rating criteria: Relevance = Relevance of the QI for the quality of home care; Potential of influence = The potential to influence the outcome measured by the QI through actions of healthcare providers; Practicability = Reliability of the coding of RAI-HC items used for QI calculation.

## Discussion

### Main findings

The study showed that the majority of RAI-HC QIs were rated to be relevant to measure quality of home care in Switzerland, irrespective of the stakeholder perspective in the consensus process. However, with regard to the potential to influence outcomes measured by the QIs, the two stakeholder perspectives resulted in different evaluations. While the experts rated 18 of 43 RAI-HC QIs as appropriate with respect to their potential of influence from a public health perspective, home care nurses rated only seven of these 18 QIs as appropriate from a healthcare providers' perspective.

### Selected quality indicators

The 18 QIs considered appropriate by the Swiss experts are multidimensional in scope and cover both physical and psychological health, as well as different functions. They measure clinical (e.g. pain, dehydration, bladder continence, skin ulcer), psychosocial (e.g. social isolation, informal caregiver distress), functional (e.g. locomotion, independency), as well as service use aspects (e.g. hospitalization). Experts pointed out missing relevant quality topics such as patient satisfaction and quality of life, which currently cannot be constructed with items of the Swiss RAI-HC. The implementation of interRAI-HC, a new version of RAI-HC, in Switzerland in the next few years offers the opportunity to measure quality of life and other topics to

ensure a comprehensive quality assessment in home care. Some QIs such as falls, cognition or weight loss, which actually are relevant from a public health [34–36] and a patient perspective [37–39], were discarded by the experts due to insufficient potential of influencing the outcome by home care organizations. While such QIs may not be appropriate measures for home care quality, they still may be useful indicators to monitor the health of the home care population and to guide public health policy in Switzerland [40].

The home care nurses discarded many more QIs due to insufficient potential of influence than the experts. As a result, and in contrast to the QIs selected by the experts, the QIs rated as appropriate by the home care nurses are less multi-dimensional. Three of the seven QIs are pain-related and the others focus on skin ulcer, obstipation, dehydration, and informal caregiver distress.

## Differences in ratings

The experts rated the appropriateness of the RAI-HC QIs on the basis of scientific evidence and with focus on the healthcare system. The home care nurses, on the other hand, primarily based their ratings on their professional and practical experience. Differences between experts' and home care nurses' views can be exemplified by the QI bladder continence decline. The experts agreed that this QI is relevant for measuring the quality of home care in Switzerland, supported by scientific evidence demonstrating the substantial economic burden of urinary incontinence (UI) to patients and society [41, 42]. In addition, evidence-based guidelines for the management of UI exist. The National Institute for Health and Care Excellence (NICE) guidelines, for example, contain various non-surgical recommendations for UI such as lifestyle interventions, physical therapies (e.g. pelvic floor muscle training), or behavioral therapies which positively influence UI [43]. The home care nurses agreed on the indicator's relevance, given the high prevalence and the negative effects on the health and quality of life of home care clients [44]. However, although evidence-based guidelines exist, they rated their potential of influence as inappropriate. On the one hand, this might be linked to the observation that nurses tend to focus on the routine management of incontinence (i.e. the use of UI pads and pants) rather than proactively address the symptoms and reasons for UI [45]. On the other hand, good continence care depends on early assessment and recognition of the problem, sufficient time resources and continence knowledge (i.e. extensive training in continence care) [46, 47]. Most home care in Switzerland is provided by less qualified staff [3] and reimbursement for assessment and prevention is limited [48]. Nurses have little room for a proactive handling of a health issue not related to the home care indication.

Home care is generally characterized by the fact that nurses have less control over outcomes compared to institutional care settings such as hospitals, nursing homes or other institutional environments where nurses work [49]. The home is the inviolable domain of clients and they have a high degree of autonomy and a say in if and how interventions will be implemented [49, 50]. Their preferences and actions can conflict with care standards, which can be illustrated by the QI inconsistent drug intake. Ellenbecker et al. [49] pointed out that inconsistent drug intake in home care can hardly be influenced if clients choose to take the medication at irregular times, despite nurses' advice on the importance of a regular medication schedule. In addition, Horrocks et al. [51] indicated that, in contrast to institutional care in hospitals or nursing homes, home care nurses are not in a position to continuously oversee clients in order to ensure compliance with best practice interventions. Moreover, it can be challenging that informal caregivers, over whom nurses have no authority, provide medical care to clients. Despite the good intentions of informal caregivers, inadequate knowledge and skills can unintentionally harm clients [52]. Finally, home care nurses work alongside various healthcare

professionals (e.g. primary care physicians, physiotherapists, occupational therapists, psycho-social service providers) and are often not solely accountable for the quality of care. Poor quality may be due to insufficient inter- and intra-professional collaboration and communication leading to mismanagement of coordinated services [49]. Research shows that collaboration and communication between healthcare professionals have an impact on the provision of healthcare and on patient outcomes [53, 54].

## Implications for further development

Quality of health care is multidimensional and QIs can be related to different dimensions such as structure, process or outcome of care [14, 55]. Multidimensionality makes it challenging to develop a set of QIs that measure quality of home care comprehensively. The 18 QIs selected by the experts as appropriate from a public health perspective reflect a wide range of QIs for measuring quality of home care in Switzerland, but mainly include outcome measures. The lack of indicators measuring processes of care was criticized by the experts and proposed as an additional quality area.

Additional process QIs would allow a better picture of home care quality, respectively of the care provided. The use of process QIs offers several advantages for home care organizations, but also for policy-makers. They are relatively easy to measure and interpret, related to what providers or nurses do (actionable), and directly point to areas that need to be improved [14, 56]. To fulfill their purpose, process measures need to be based on strong clinical evidence showing positive associations between implementation of state-of-the-art care processes and clinical outcomes (process-outcome link). Ideally, process QIs are generated from evidence-based clinical practice guidelines [17, 57, 58]. Many such practice guidelines exist [59, 60], but are not necessarily developed for home care. Further research into home care specific guidelines to support best practices and the development of process QIs is recommended. Such a development should also consider QIs that measure coordination processes in home care and the impact of inter- and intra-professional collaborations on home care clients outcomes [7, 53].

In addition to the different dimension of quality, the intended use of QIs should be considered when developing or deciding on QIs. So far, home care QIs in Switzerland were only used for internal quality management, the federal authorities aim at using QIs for monitoring and benchmarking national care quality and care impact. The results of our study provide a list of 18 QIs rated as appropriate from a public health perspective, thus relevant for federal authorities. However, from the practice perspective, only 7 QIs were considered influenceable, limiting the acceptance of the other QIs for national level use. Therefore, further research is needed to explore which QIs capture the impact of high quality services in homes care. Further, for public reporting, the development of a comprehensive risk adjustment for a fair quality comparison of home care organizations is needed [61].

## Strength and limitations

To our knowledge, this is the first study assessing the appropriateness of RAI-HC QIs to measure quality of home care systematically and comparing their appropriateness from a public health and healthcare providers' perspective. These perspectives reflect different aims in measuring QIs in home care. The perspective of home care nurses is less frequently taken into account in QI research even though nurses play a key role in providing care and quality improvement. A major strength of this study was that the selection of the QIs was based on a recent systematic literature review [22] as well as the multidisciplinary expert panel

representing the three official language regions of Switzerland in order to take cultural differences between the regions into account.

One substantively notable limitation is that some stakeholders such as patients, healthcare insurers and primary care physicians were not included in the evaluation process. Also, it is conceivable, that a different selection of experts might have come to a different set of QIs. Moreover, the selection of QIs by the home care nurses could have been different, had they received the complete or a different set of QIs. Another limitation is that our study results may not be generalizable to other countries. Even though RAI-HC is an international instrument and QIs can be operationalized in various countries, cultural and contextual differences limit the generalizability of our current findings [62].

## Conclusions

The study underlines the importance of evaluating the appropriateness of RAI-HC QIs to measure the quality of home care in Switzerland from different stakeholder perspectives. While both stakeholder groups, experts and home care nurses, showed a high agreement on the relevance of RAI-HC QIs, we found heterogeneous results with regard to the potential of influence QIs. Differences can be explained by different perspectives, population- vs. patient-level, and the experienced limited scope of action and influence on clients' outcomes by home care nurses. They indicate the necessity to specify the limitations and purpose, public or individual health, of QIs in a given context. As home care quality is multidimensional, a comprehensive quality assessment requires a certain number of QIs. The seven QI rated as appropriate by home care nurses would not suffice, while the 18 resulting from the expert rating cover a wide scope. Adding process QIs and additional QIs on patient satisfaction or quality of life would improve the overall quality assessment. The findings can help Swiss policy-makers, healthcare managers and home care organizations in choosing appropriate QIs for their intended use.

## Supporting information

**S1 File. CREDES checklist.**
(DOCX)

## Acknowledgments

The authors would like to thank Cornelis Kooijman and Esther Bättig from the Swiss Association of Home Care Organizations (Spitex Schweiz) for their support, and the experts and home care nurses who participated in the panel and focus groups for their interest and time.

The study relates to ongoing work by the HCD (HomeCareData) Research Group. The HCD Research Group consists of:

Institute of Health Sciences, ZHAW: Julia Dratva, René Schaffert, Aylin Wagner

Winterthur Institute of Health Economics, ZHAW: Eva Hollenstein, Florian Liberatore, Sarah Schmelzer

Swiss Health Observatory (OBSAN): Laure Dutoit, Sonia Pellegrini

Institute of Social and Preventive Medicine, University of Bern: Adrian Spoerri, Andreas Boss

The lead author of the HCD Research Group is Prof. Dr. med. Julia Dratva (julia.dratva@z-haw.ch).

## Author Contributions

**Conceptualization:** Aylin Wagner, Franziska Zúñiga, Peter Rüesch, René Schaffert, Julia Dratva.

**Funding acquisition:** Peter Rüesch, Julia Dratva.

**Investigation:** Aylin Wagner, Franziska Zúñiga, René Schaffert, Julia Dratva.

**Methodology:** Aylin Wagner, Franziska Zúñiga, René Schaffert, Julia Dratva.

**Project administration:** Aylin Wagner, René Schaffert, Julia Dratva.

**Supervision:** René Schaffert, Julia Dratva.

**Writing – original draft:** Aylin Wagner.

**Writing – review & editing:** Aylin Wagner, Franziska Zúñiga, Peter Rüesch, René Schaffert, Julia Dratva.

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
