## [Decision Letter · Decision Letter 0]

2 Oct 2020

PONE-D-20-11808

Selecting home care quality indicators based on the Resident Assessment Instrument-Home Care (RAI-HC) for Switzerland: a public health and healthcare providers' perspective

PLOS ONE

Dear Dr. Wagner,

Thank you for submitting your manuscript to PLOS ONE. After careful consideration, we feel that it has merit but does not fully meet PLOS ONE’s publication criteria as it currently stands. Therefore, we invite you to submit a revised version of the manuscript that addresses the points raised during the review process.

We look forward to receiving your revised manuscript.

Kind regards,

Valérie Pittet, PhD

Academic Editor

PLOS ONE

2. In your Methods section, please provide additional information about the participant recruitment method and the demographic details of your participants. Please ensure you have provided sufficient details to replicate the analyses such as: a) the recruitment date range (month and year), b) a description of any inclusion/exclusion criteria that were applied to participant recruitment, c) a table of relevant demographic details, d) a statement as to whether your sample can be considered representative of a larger population, e) a description of how participants were recruited, and f) descriptions of where participants were recruited and where the research took place.

3. One of the noted authors is a group or consortium [HCD Research Group]. In addition to naming the author group, please list the individual authors and affiliations within this group in the acknowledgments section of your manuscript. Please also indicate clearly a lead author for this group along with a contact email address.

4. We noted in your submission details that a portion of your manuscript may have been presented or published elsewhere.

[Our manuscript relates to our systematic review on RAI-HC quality indicators performed in the context of the same study (forthcoming, Wagner et al., BMC Health Serv Res). In the submitted manuscript, the RAI-HC quality indicators identified in our systematic review were further analysed by having experts and health care providers evaluate their appropriateness of measuring quality of home care in Switzerland. The papers report on two separate sub-studies and do not overlap. The pending paper does not constitute dual publication. A copy of the paper reporting on our systematic literature review has been uploaded. ]

Reviewers' comments:

Reviewer's Responses to Questions

**Comments to the Author**

1. Is the manuscript technically sound, and do the data support the conclusions?

Reviewer #1: Yes

Reviewer #2: Yes

2. Has the statistical analysis been performed appropriately and rigorously? 

Reviewer #1: Yes

Reviewer #2: Yes

3. Have the authors made all data underlying the findings in their manuscript fully available?

Reviewer #1: Yes

Reviewer #2: No

4. Is the manuscript presented in an intelligible fashion and written in standard English?

Reviewer #1: Yes

Reviewer #2: Yes

5. Review Comments to the Author

Reviewer #1: When you mention the different language regions of Switzerland, could you provide a little more elaboration on that? I am guessing that the surveys were pre-tested for these various populations, correct?

On lines 148-149, it appears a little confusing as to what you mean by QI numerators and denominators. Please elaborate further.

On line 162, change to “analysis was carried out…”

Around line 166, I would think that a greater elaboration on the “disagreement index” might be warranted.

On line 176 (and perhaps other places), you use the word “criterium”, which refers to a cycling race. Is it possible you mean “criterion”? It also shows up on line 227 and other places as well.

On line 206, I believe that you meant to say “rationale”.

On line 211, replace with “relevance, potential of influence, practicability, and consensus”.

Reviewer #2: This paper performed a series of discussion groups that asked health care professionals from Switzerland to evaluate a set of quality indicators for home care. Through these discussion groups, participants were asked to select quality indicators that they felt were appropriate for home care in Switzerland based on their expertise and experience. The results found 18 out of 43 indicators available from literature review. The results of ratings used to select the final indicators were reported. Some discussions were provided regarding why these were selected.

However, in its current form, the manuscript is not ready for publication. I am not suggesting any new experiment, but a major revision is required. I believe the manuscript lacks depth in the following areas: context of home care in Switzerland (comments #1-3), rationales for chosen methods (comment #4), lack of reporting of qualitative results or observations from discussion groups (comment #5), and lack of interpretation and further synthesis of the results (comment #6).

Major comments

1. In the Background section, I think the authors should elaborate more on the definition of home care. In addition, I think readers would appreciate some description of how home care in Switzerland is organized, funded, staffed, and delivered to fully understand to what extent the results can be generalizable beyond Switzerland.

2. The authors made some comments regarding home care versus institutional care throughout the manuscript. Therefore, it may also be helpful to define what you mean by institutional care. Did you mean inpatient acute care or nursing home or some other types of services?

3. I believe the discussions need to speak to potential usage of these QIs in Switzerland's home care. For example, will there be incentive, bonus, or penalty attached to each QI? If not, how are performance being rewarded or penalized? In addition, I was very curious about how home care performance are currently being evaluated, if any (which should also be included in the Introduction).

4. In phase 2, the health care providers were only given 18 QIs from phase 1, and not the full list. First, I could not find a rationale for this in the Methods section. Second, is it not a limitation if the health care providers did not get to review the full list? Could more QI be selected otherwise?

5. The results section reported mostly quantitative results. I am curious why the qualitative results from the focus groups, panel meetings, etc. were not included? Were they any themes that the participants discussed beyond the quantitative measures? For example, nurses mentioned lack of control over taking medications in the discussions. Were there others? Reporting these will help your discussion on rationale of ratings.

6. I am not sure why urinary incontinence was the focus in the discussions, and was the only example mentioned in the discussion. Other QI's, such as Hospitalization or falls due to difficulty in locomotion, can arguably be expensive for the health care system as well.

My suggestions for the discussions:

6.1 You have a set of chosen QI's. Let's first summarize the themes of these QI's. For example, it was mentioned that 7 were pain related. What about the others?

6.2 Similarly which themes or groups were not included?

6.3 This may help set you up to explain why some QI themes were included and some weren't.

6.4 Why each theme, provide some rationales for selection or rejection.

6.5 Everything should tie back to the context of home care in Switzerland of why something makes sense and others did not.

Minor comments

7. "The perspective of healthcare providers, for example, focus primarily on the care provided by practitioners to individual patients." on lines 69-70 need a citation.

8. "Literatur" misspelling Figure 1.

6. PLOS authors have the option to publish the peer review history of their article (what does this mean?). If published, this will include your full peer review and any attached files.

Reviewer #1: No

Reviewer #2: No

---

## [Author Response · Author response to Decision Letter 0]

11 Nov 2020

Author’s response to reviews

Title: Selecting home care quality indicators based on the Resident Assessment Instrument-Home Care (RAI-HC) for Switzerland: a public health and healthcare providers' perspective

Authors:

Aylin Wagner (aylin.wagner@zhaw.ch)

Franziska Zúñiga (franziska.zuniga@unibas.ch)

Peter Rüesch (peter.ruuesch@zhaw.ch)

René Schaffert (rene.schaffert@zhaw.ch)

Julia Dratva (julia.dratva@zhaw.ch)

Version: 1 Date: 11 Nov 2020

Dear Dr. Pittet, dear reviewers

Thank you for the valuable and constructive comments and for the opportunity to revise our submission to PLOS ONE. We have revised our manuscript following the academic editor’s and reviewers’ comments and attempted to address all questions and comments raised by the reviewers. The changes and revisions made to the manuscript have been highlighted in yellow. 

Please find our point to point comments below. 

Yours sincerely,

Aylin Wagner

Response to reviewers' comments

Reviewer #1: 

• When you mention the different language regions of Switzerland, could you provide a little more elaboration on that? I am guessing that the surveys were pre-tested for these various populations, correct?

Response: We are happy to elaborate some more on the specific Swiss situation. Switzerland comprises three main language regions with different local cultures. There are no fundamental differences in the structures of health care between the different language regions, apart from cantonal variations due to their political self-determination [1]. However, subtle variations in the different language parts regarding aspects of health care exist due to differing beliefs, norms, expectations, and matters of preference [1]. We recruited experts and home care nurses from the three major language regions of Switzerland (German, French and Italian speaking) in order to take cultural differences into account and to assess the appropriateness of the RAI-HC QIs throughout the country. We revised the section to improve clarity (it appears on page 7, line 159-161):

“The multidisciplinary expert panel consisted of 14 members from three language regions of Switzerland in order to take cultural differences into account and to examine the appropriateness of RAI-HC QIs throughout the country.“ (Methods, page 7, line 159-161)

Switzerland is multilingual. The experts in our study had at least a passive knowledge of another national language. Experts, whose mother tongue was not German but French or Italian, had very good passive language skills in German. Therefore, it was not necessary to translate the survey into another language and pre-test it. The expert panel meeting was offered as a multilingual event, i.e. the experts could speak their first language, which is common practice in expert meetings in Switzerland. We revised the sections accordingly:

“Experts whose mother tongue was not German had very good passive language skills in German.” (Methods, page 7, line 162-163)

“The panel members attended a face-to-face multilingual meeting (i.e. the experts could talk in their first language), led by an experienced moderator.” (Methods, page 9, line 211-212)

The focus groups with the home care nurses were held in German and French, respectively. Technical terms in the QI definitions were already available in French as the Swiss version of RAI-HC was professionally translated into French several years ago. Further, focus group study documents were professionally translated. Participants from the Italian-speaking part had very good knowledge of French and therefore took part in the French-speaking focus group. Again, a common practice in Switzerland to resolve multilinguality. We added information on the language of the focus group to improve clarity (it appears on page 10, line 223-226):

“The QIs selected in the expert panel were further evaluated based on the Nominal-Group-Technique (NGT) [28] in three focus groups with registered nurses from Swiss home care organizations situated in the three main language regions. The documents for the focus groups were professionally translated from German into French, and focus groups were held in the respective language.” (Methods, page 10, line 223-226)

• On lines 148-149, it appears a little confusing as to what you mean by QI numerators and denominators. Please elaborate further.

Response: We agree that understanding can be improved and added information in the background section on QI calculation with a numerator and denominator to improve clarity (it appears on page 5, line 98-101):

“The RAI-HC QIs are constructed as proportions or percentages, expressed by a fractional calculation with numerator (number of clients with a particular outcome) and denominator (number of clients at risk for the outcome and not otherwise excluded from the QI) [5,21].” (Background, page 5, line 98-101)

• On line 162, change to “analysis was carried out…”

Response: We agree and have rephrased this sentence accordingly (it now appears on page 8, line 188):

“The analysis was carried out in accordance with the RAND/UCLA Appropriateness Method user's manual [26].” (Methods, page 8, line 188)

• Around line 166, I would think that a greater elaboration on the “disagreement index” might be warranted.

Response: We agree and have added further information on the Disagreement Index (DI) to improve clarity (it now appears on page 9, line 192-195). Detail information on the calculation of the IPR and IPRAS can be found in “The RAND/UCLA Appropriateness Method User’s Manual” (available online) [2]. This reference is also in the manuscript.

“The DI is based on the dispersion of the distribution (interpercentile range, IPR) and symmetry (interpercentile range adjusted for symmetry, IPRAS) of the ratings on the 9-point scale and is calculated with the formula: IPR (difference between 30th and 70th percentile) divided by the IPRAS.” (Methods, page 9, line 192-195)

• On line 176 (and perhaps other places), you use the word “criterium”, which refers to a cycling race. Is it possible you mean “criterion”? It also shows up on line 227 and other places as well.

Response: This is a misspelling. We have corrected it in our manuscript:

“All of the related QIs were classified as uncertain in the first rating round, should one of the QIs in question be classified as uncertain or discarded, and were thus discussed and re-rated with respect to the criterion potential of influence in the panel meeting.” (Methods, page 9, line 202)

“The majority of QIs were discarded because of inappropriate rating results with respect to the criterion potential of influence.” (Results, page 11, line 260)

“If one of three QIs related to the same health outcome were classified as uncertain or discarded in rating round 1, then all three QIs were classified as uncertain (regardless of the actual rating result) and were re-rated according to the criterion potential of influence in rating round 2.” (Table 4, page, 14, line 265-267)

• On line 206, I believe that you meant to say “rationale”.

Response: This is a misspelling. We have corrected it in our manuscript:

“Between the two rating rounds, the ratings for each QI were collected and followed by a discussion in which participants described the rationale of their ratings.” (Methods, page 10, line 240-241)

• On line 211, replace with “relevance, potential of influence, practicability, and consensus”.

Response: We agree and have rephrased this sentence accordingly:

“As in the expert panel, median scores and level of agreement (DI) among participants were calculated and the rule for QI selection was applied (see Table 3), i.e. only QIs with appropriate median scores of relevance, potential of influence, practicability, and consensus, were selected.” (Methods, page 11, line 243-245)

Reviewer #2: 

• This paper performed a series of discussion groups that asked health care professionals from Switzerland to evaluate a set of quality indicators for home care. Through these discussion groups, participants were asked to select quality indicators that they felt were appropriate for home care in Switzerland based on their expertise and experience. The results found 18 out of 43 indicators available from literature review. The results of ratings used to select the final indicators were reported. Some discussions were provided regarding why these were selected.

However, in its current form, the manuscript is not ready for publication. I am not suggesting any new experiment, but a major revision is required. I believe the manuscript lacks depth in the following areas: context of home care in Switzerland (comments #1-3), rationales for chosen methods (comment #4), lack of reporting of qualitative results or observations from discussion groups (comment #5), and lack of interpretation and further synthesis of the results (comment #6).

Response: We would like to thank the reviewer for the time spent reviewing our manuscript and providing useful and detailed comments to improve our manuscript. We have studied the suggestions carefully. Please find our point to point comments below.

Major comments

1. In the Background section, I think the authors should elaborate more on the definition of home care. In addition, I think readers would appreciate some description of how home care in Switzerland is organized, funded, staffed, and delivered to fully understand to what extent the results can be generalizable beyond Switzerland.

Response: Thank you for pointing out the need for more background information. We have added a paragraph in the background chapter with information about home care services in Switzerland in order to give the reader a better understanding of the setting (it appears on page 4, line 64-75):

“Home care services in Switzerland are intended for people of all age groups in need of care or assistance at home and are run by profit and non-profit home care organizations as well as independent nurses. Four‐fifths of Swiss home care clients receive services from non-profit home care organizations [3]. The range of services offered by home care organizations includes nursing care and domestic tasks [4]. The compulsory health insurance pays for care services prescribed by general practitioners (GPs) but not for domestic tasks. The organizational structure of home care is highly decentralized and ultimately reflects the federal political structure of Switzerland. Home care plays an important role in managing interfaces between primary care, acute care, long-term care and mental health services [5] and is characterized by interprofessional collaboration, i.e. nurses, GPs and other health care providers (e.g. pharmacists, physiotherapist) work together to provide a wide range of services to clients [6,7].” (Background, page 4, line 64-75)

2. The authors made some comments regarding home care versus institutional care throughout the manuscript. Therefore, it may also be helpful to define what you mean by institutional care. Did you mean inpatient acute care or nursing home or some other types of services?

Response: Institutional care refers to a living environment designed to meet the functional, medical, personal, social and housing needs of people with physical or mental disabilities. In our paper, we used the term, institutional care, according to Ellenbecker et al. [3] and Horrocks et al. [4] meaning hospitals and nursing homes or other institutional environments where nurses work. We rephrased the sentences to improve clarity:

“Home care is generally characterized by the fact that nurses have less control over outcomes compared to institutional care settings such as hospitals, nursing homes or other institutional environments where nurses work [49].” (Discussion, page 17, line 349-351)

“In addition, Horrocks et al. [51] indicated that, in contrast to institutional care in hospitals or nursing homes, home care nurses are not in a position to continuously oversee clients in order to ensure compliance with best practice interventions.” (Discussion, page 17-18, line 356-357)

3. I believe the discussions need to speak to potential usage of these QIs in Switzerland's home care. For example, will there be incentive, bonus, or penalty attached to each QI? If not, how are performance being rewarded or penalized? In addition, I was very curious about how home care performance are currently being evaluated, if any (which should also be included in the Introduction).

Response: We thank the reviewer for this valuable comment and are happy to provide further information on the usage. The Swiss RAI-HC QIs were developed for internal quality management of home care organizations. However, since 2016 there is a legal basis in Switzerland that obliges health care providers to report data on QIs to the respective federal authorities. Currently this law is not being implemented for home care. The Federal Office of Public Health (FOPH) will define which QIs will be collected at the national level in near future. A FOPH representative is in the advisory board of the present study and our findings will support the FOPH in the selection of suitable national QIs. We added information regarding the current use of the Swiss RAI-HC QIs in the background chapter (it appears on page 5, line 110-117):

“The Swiss RAI-HC QIs have so far only been used for internal quality management in non-profit home care organizations. To date, in Switzerland, RAI-HC QIs (or any home care QIs) are not reported and there are no national standards for home care [8]. However, there is a legal basis that obliges home care organizations to report data on QIs to the respective federal authorities with a goal of public reporting [25]. Currently this law is not being implemented due to lack of knowledge which QIs are the most appropriate. The Federal Office of Public Health (FOPH) will define which QIs will be collected at the national level in near future. No incentives will be linked to the QI reporting.” (Background, page 5, line 110-117)

In the conclusion chapter (page 20, line 427-428) we stated that our findings can help Swiss policy-makers, healthcare managers and home care organizations in choosing appropriate QIs for their intended use. We now elaborate more on the potential use (“Implications for further development”) in the discussion chapter (it appears on page 19, line 388-396):

“In addition to the different dimension of quality, the intended use of QIs should be considered when developing or deciding on QIs. So far, home care QIs in Switzerland were only used for internal quality management, the federal authorities aim at using QIs for monitoring and benchmarking national care quality and care impact. The results of our study provide a list of 18 QIs rated as appropriate from a public health perspective, thus relevant for federal authorities. However, from the practice perspective, only 7 QIs were considered influenceable, limiting the acceptance of the other QIs for national level use. Therefore, further research is needed to explore which QIs capture the impact of high quality services in homes care. Further, for public reporting, the development of a comprehensive risk adjustment for a fair quality comparison of home care organizations is needed [61].” (Discussion, page 19, line 388-396)

4. In phase 2, the health care providers were only given 18 QIs from phase 1, and not the full list. First, I could not find a rationale for this in the Methods section. Second, is it not a limitation if the health care providers did not get to review the full list? Could more QI be selected otherwise?

Response: Both methods used in our study represent systematic consensus techniques [5] and provide quantitative results on which QIs can be selected or discarded. Given the study question, we aimed at defining the QIs appropriate from a public health perspective using the RAND/UCLA Appropriateness Method (RAM) and then evaluating these from a practical perspective using the Nominal-Group-Technique (NGT). Both techniques were chosen for specific reasons:

RAM: The aim of the first phase was to assess the appropriateness of RAI-HC QIs from a public health perspective. The RAM is an adequate method to identify the collective opinion of experts, is state-of-the-art technique and has been incorporated into a comprehensive approach for the development of QIs in various contexts [6–10] (page 7, line 152). 

NGT: The aim of the second phase was to further evaluate the QIs rated appropriate by the experts from a healthcare providers’ perspective. The NGT was a suitable method for this further evaluation since the list of QIs was reduced (from 43 to 18 QIs). The NGT is a highly structured process in which a reasonable number of indicators can be discussed and rated. Moreover, the NGT is a method in which participants are brought together for both rating rounds (in contrast to the RAM). Since the home care nurses had no expertise with regard to the construction and use of QIs, the method allowed the research team to provide oral explanations on the QIs and to help the participants with uncertainties during the rating process. We added the rationale of the method to improve clarity:

“The QIs selected in the expert panel were further evaluated based on the Nominal-Group-Technique (NGT) [28] in three focus groups with registered nurses from Swiss home care organizations situated in the three main language regions.” (Methods, page, 10 line 223-225)

“The NGT [28] is a structured consensus process and is based on a strongly structured meeting in which individual and group work alternate” (Methods, page 10, line 233-234)

“The method allowed the research team to provide oral explanations on the QIs and to help participants in case of uncertainties during the rating process, as the home care nurses had no expertise in QI construction and use.” (Methods, page 10, line 236-238)

Having provided the reduced list can be seen as a limitation, the full set may have led to a different selection of QIs by nurses. We have included this point as a limitation of our study (it appears on page 19, line 409-411):

“Moreover, the selection of QIs by the home care nurses could have been different, had they received the complete or a different set of QIs.” (Discussion, page 19 , line 409-411)

5. The results section reported mostly quantitative results. I am curious why the qualitative results from the focus groups, panel meetings, etc. were not included? Were they any themes that the participants discussed beyond the quantitative measures? For example, nurses mentioned lack of control over taking medications in the discussions. Were there others? Reporting these will help your discussion on rationale of ratings.

Response: Presenting qualitative results are outside the scope of this study. Our study designs and methodologies are strictly quantitative (consensus surveys). Both methods applied (RAM and NGT) do not include the analysis of the discussions among participants. 

The expert panel meeting and focus groups were rigorously structured meetings with a fixed time window to discuss and rate the QIs, i.e. the participants only discussed the QIs (quantitative measures) based on the rating criteria. 

So unfortunately, we cannot add any qualitative results, but we agree that this would be very interesting.

6. I am not sure why urinary incontinence was the focus in the discussions, and was the only example mentioned in the discussion. Other QI's, such as Hospitalization or falls due to difficulty in locomotion, can arguably be expensive for the health care system as well.

Response: We agree that there are many other QIs among the 43 QIs discussed in the first phase of importance for the health care system and clients. We have chosen urinary incontinence only as an example because it nicely illustrates the discrepancy between the public health and health care providers’ perspective, as explained in the paper. We changed the text to make clearer that we are only providing an example (it appears on page 17, line 331-332):

“Differences between experts’ and home care nurses’ views can be exemplified by the QI bladder continence decline.” (Discussion, page 17, line 331-332)

My suggestions for the discussions:

6.1 You have a set of chosen QI's. Let's first summarize the themes of these QI's. For example, it was mentioned that 7 were pain related. What about the others?

Response: Thank you, we are happy to provide more information on the QI themes.

Table 4 gives an overview of all QIs. The discussion chapter includes a summary of themes of the QIs. On page 16, line 310-314, we summarize the areas/themes (with QI examples) covered by the 18 QIs selected by the experts and on page 16, line 325-326, we mention that three of the seven QIs selected by the home care nurses are pain-related and list the other four QIs. We have moved the sub-chapter "Selected quality indicators" after the chapter "Main results", so the information about the selected QIs appears at the beginning of the discussion chapter. We also provided further information on the currently existing RAI-HC QIs and the areas they cover in the background chapter (it appears on page 5, line 101-105): 

“A systematic review showed that currently 48 RAI-HC QIs exist [22]. These QIs cover different areas relevant to home care, focusing on functional (e.g. activities of daily living, cognition, communication, hearing, eyesight), clinical (e.g. bladder incontinence, bowel incontinence, skin ulcer, mouth problems, falls, weight, mood, pain), social (informal caregivers, social isolation), and service use aspects (flu vaccination, hospitalization) [22].” (Background, page 5, line 101-105)

6.2 Similarly which themes or groups were not included?

Response: On page 16, line 318, we mention some of the excluded themes (e.g. falls, cognition, weight loss). Unfortunately, it was not possible to list all excluded QIs by groups, because many QIs cannot be combined into groups, however, tables 4 and 5 in the results chapter give an overview of rating results and inclusion or exclusion of QIs.

6.3 This may help set you up to explain why some QI themes were included and some weren't.

Response: As mentioned earlier, the aim of our study was to quantitatively evaluate the appropriateness of QIs for the Swiss context. QIs were selected/rejected based on median scores and consensus as described in table 3 in the method chapter (Table 3. Classification rules of quality indicators, page 9). QIs were discarded because of low median scores or lack of consensus in the corresponding rating criteria. Both experts and home care nurses were most critical of the criterion "potential for influence" and it was therefore the main reason why QIs were rejected. 

6.4 Why each theme, provide some rationales for selection or rejection.

Response: The title “Rationale of ratings” in the discussion chapter is misleading and raises the expectation that we provide the rationale of the rating for each QI. We have therefore changed the title of this chapter (it appears now on page 17, line 328): 

“Differences in ratings” (Discussion, page 17, line 328)

Observations from the expert panel meeting and focus groups showed that the criterion ratings were heterogenous and often unique for each QIs or quality area. To exemplify the difference in ratings and to link the ratings to the home care context, we used the QI urinary incontinence and inconsistent drug intake. Based on these examples, and the scientific literature, we showed why the potential of influence (main reason for rejection of QIs) is limited in the home care setting. Unfortunately, we cannot discuss rationales of ratings for each QIs or area because it would go beyond the scope of the discussion chapter. 

6.5 Everything should tie back to the context of home care in Switzerland of why something makes sense and others did not.

Response: We agree that our results must be discussed in the Swiss context of home care and we added some more thoughts on the specific Swiss situation in the background and discussion chapter of our paper: 

“Home care services in Switzerland are intended for people of all age groups in need of care or assistance at home and are run by profit and non-profit home care organizations as well as independent nurses. Four‐fifths of Swiss home care clients receive services from non-profit home care organizations [3]. The range of services offered by home care organizations includes nursing care and domestic tasks [4]. The compulsory health insurance pays for care services prescribed by general practitioners (GPs) but not for domestic tasks. The organizational structure of home care is highly decentralized and ultimately reflects the federal political structure of Switzerland. Home care plays an important role in managing interfaces between primary care, acute care, long-term care and mental health services [5] and is characterized by interprofessional collaboration, i.e. nurses, GPs and other health care providers (e.g. pharmacists, physiotherapist) work together to provide a wide range of services to clients [6,7].” (Background, page 4, line 64-75)

“The Swiss RAI-HC QIs have so far only been used for internal quality management in non-profit home care organizations. To date, in Switzerland, RAI-HC QIs (or any home care QIs) are not reported and there are no national standards for home care [8]. However, there is a legal basis that obliges home care organizations to report data on QIs to the respective federal authorities with a goal of public reporting [25]. Currently this law is not being implemented due to lack of knowledge which QIs are the most appropriate. The Federal Office of Public Health (FOPH) will define which QIs will be collected at the national level in near future. No incentives will be linked to the QI reporting.” (Background, page 5, line 110-117)

“Most home care in Switzerland is provided by less qualified staff [3] and reimbursement for assessment and prevention is limited [48]. Nurses have little room for a proactive handling of a health issue not related to the home care indication.” (Discussion, page 17, line 345-348)

“Quality of health care is multidimensional and QIs can be related to different dimensions such as structure, process or outcome of care [14,55]. Multidimensionality makes it challenging to develop a set of QIs that measure quality of home care comprehensively. The 18 QIs selected by the experts as appropriate from a public health perspective reflect a wide range of QIs for measuring quality of home care in Switzerland, but mainly include outcome measures. The lack of indicators measuring processes of care was criticized by the experts and proposed as an additional quality area.” (Discussion, page 18, line 370-375)

“In addition to the different dimension of quality, the intended use of QIs should be considered when developing or deciding on QIs. So far, home care QIs in Switzerland were only used for internal quality management, the federal authorities aim at using QIs for monitoring and benchmarking national care quality and care impact. The results of our study provide a list of 18 QIs rated as appropriate from a public health perspective, thus relevant for federal authorities. However, from the practice perspective, only 7 QIs were considered influenceable, limiting the acceptance of the other QIs for national level use. Therefore, further research is needed to explore which QIs capture the impact of high quality services in homes care. Further, for public reporting, the development of a comprehensive risk adjustment for a fair quality comparison of home care organizations is needed [61].” (Discussion, page 19, line 388-396)

Minor comments

7. "The perspective of healthcare providers, for example, focus primarily on the care provided by practitioners to individual patients." on lines 69-70 need a citation.

Response: We agree and added citations (it now appears on page 4, line7 9-80):

“The perspective of healthcare providers, for example, focuses primarily on the care provided to individual patients [10,11].” (Background, page 4, line 81-82)

8. "Literatur" misspelling Figure 1.

Response: We fixed the spelling error in Figure 1.

 

References

1. Jenni OG, Sennhauser FH. Child Health Care in Switzerland. The Journal of Pediatrics. 2016;177: S203–S212. doi:10.1016/j.jpeds.2016.04.056

2. Fitch K, Bernstein SJ, Aguilar MD, Burnand B, LaCalle JR, Lazaro P, et al., editors. The RAND/UCLA Appropriateness Method User’s Manual. Santa Monica, Calif.: RAND Corporation; 2001. 

3. Ellenbecker CH, Samia L, Cushman MJ, Alster K. Patient Safety and Quality in Home Health Care. In: Hughes RG, editor. Patient Safety and Quality: An Evidence-Based Handbook for Nurses. Rockville, MD: Agency for Healthcare Research and Quality; 2008. 

4. Horrocks S, Pollard K, Duncan L, Petsoulas C, Gibbard E, Cook J, et al. Measuring quality in community nursing: a mixed-methods study. Health Serv Deliv Res. 2018;6: 1–132. doi:10.3310/hsdr06180

5. Campbell S, Braspenning J, Hutchinson A, Marshall M. Research methods used in developing and applying quality indicators in primary care. Qual Saf Health Care. 2002;11: 358–364. doi:10.1136/qhc.11.4.358

6. Koenders N, van den Heuvel S, Bloemen S, van der Wees PJ, Hoogeboom TJ. Development of a longlist of healthcare quality indicators for physical activity of patients during hospital stay: a modified RAND Delphi study. BMJ Open. 2019;9: e032208. doi:10.1136/bmjopen-2019-032208

7. Yajima N, Tsujimoto Y, Fukuma S, Sada K, Shimizu S, Niihata K, et al. The development of quality indicators for systemic lupus erythematosus using electronic health data: A modified RAND appropriateness method. Modern Rheumatology. 2019; 1–7. doi:10.1080/14397595.2019.1621419

8. Spackman E, Clement F, Allan GM, Bell CM, Bjerre LM, Blackburn DF, et al. Developing key performance indicators for prescription medication systems. Tang Y, editor. PLoS ONE. 2019;14: e0210794. doi:10.1371/journal.pone.0210794

9. Bitton A, Vutcovici M, Lytvyak E, Kachan N, Bressler B, Jones J, et al. Selection of Quality Indicators in IBD: Integrating Physician and Patient Perspectives. Inflammatory Bowel Diseases. 2019;25: 403–409. doi:10.1093/ibd/izy259

10. McCorry NK, O’Connor S, Leemans K, Coast J, Donnelly M, Finucane A, et al. Quality indicators for Palliative Day Services: A modified Delphi study. Palliat Med. 2019;33: 197–205. doi:10.1177/0269216318810601

---

## [Decision Letter · Decision Letter 1]

14 Dec 2020

Selecting home care quality indicators based on the Resident Assessment Instrument-Home Care (RAI-HC) for Switzerland: a public health and healthcare providers' perspective

PONE-D-20-11808R1

Dear Dr. Wagner,

We’re pleased to inform you that your manuscript has been judged scientifically suitable for publication and will be formally accepted for publication once it meets all outstanding technical requirements.

Kind regards,

Valérie Pittet, PhD

Academic Editor

PLOS ONE

Reviewers' comments:

Reviewer's Responses to Questions

**Comments to the Author**

1. If the authors have adequately addressed your comments raised in a previous round of review and you feel that this manuscript is now acceptable for publication, you may indicate that here to bypass the “Comments to the Author” section, enter your conflict of interest statement in the “Confidential to Editor” section, and submit your "Accept" recommendation.

Reviewer #1: All comments have been addressed

2. Is the manuscript technically sound, and do the data support the conclusions?

Reviewer #1: Yes

3. Has the statistical analysis been performed appropriately and rigorously? 

Reviewer #1: Yes

4. Have the authors made all data underlying the findings in their manuscript fully available?

Reviewer #1: Yes

5. Is the manuscript presented in an intelligible fashion and written in standard English?

Reviewer #1: Yes

6. Review Comments to the Author

Reviewer #1: I believe that the authors have provided the requisite changes that were asked of them by the initial reviewers. The subject matter seems important as well, and thus I believe that it will be wekk received by the PLoS ONE audience.

7. PLOS authors have the option to publish the peer review history of their article (what does this mean?). If published, this will include your full peer review and any attached files.

Reviewer #1: **Yes: **Andrew T Carswell

---

## [Editor Report · Acceptance letter]

18 Dec 2020

PONE-D-20-11808R1 

Selecting home care quality indicators based on the Resident Assessment Instrument-Home Care (RAI-HC) for Switzerland: a public health and healthcare providers' perspective 

Dear Dr. Wagner:

I'm pleased to inform you that your manuscript has been deemed suitable for publication in PLOS ONE. Congratulations! Your manuscript is now with our production department. 

Kind regards, 

on behalf of

PD Dr. Valérie Pittet 

Academic Editor

PLOS ONE